# Label-Free Surface-Enhanced Raman Spectroscopic Analysis of Proteins: Advances and Applications

**DOI:** 10.3390/ijms232213868

**Published:** 2022-11-10

**Authors:** Linjun Cai, Guilin Fang, Jinpin Tang, Qiaomei Cheng, Xiaoxia Han

**Affiliations:** 1National Engineering Laboratory for AIDS Vaccine, School of Life Science, Jilin University, Changchun 130012, China; 2State Key Laboratory of Supramolecular Structure and Materials, College of Chemistry, Jilin University, Changchun 130012, China

**Keywords:** SERS, protein, structural characterization, biomarker, cell, pathogen

## Abstract

Surface-enhanced Raman spectroscopy (SERS) is powerful for structural characterization of biomolecules under physiological condition. Owing to its high sensitivity and selectivity, SERS is useful for probing intrinsic structural information of proteins and is attracting increasing attention in biophysics, bioanalytical chemistry, and biomedicine. This review starts with a brief introduction of SERS theories and SERS methodology of protein structural characterization. SERS-active materials, related synthetic approaches, and strategies for protein-material assemblies are outlined and discussed, followed by detailed discussion of SERS spectroscopy of proteins with and without cofactors. Recent applications and advances of protein SERS in biomarker detection, cell analysis, and pathogen discrimination are then highlighted, and the spectral reproducibility and limitations are critically discussed. The review ends with a conclusion and a discussion of current challenges and perspectives of promising directions.

## 1. Introduction

Raman scattering is an inelastic scattering process in which energy shifts are involved upon light-molecule interactions [1] (Figure 1A). Vibrational frequencies are specific to a molecule’s chemical bonds, on the basis of which Raman spectroscopy can provide detailed structural information of molecules. The Raman scattering of a molecule would be significantly enhanced when the molecule is either attached to or in proximity to a nanostructured surface, known as surface-enhanced Raman scattering [2] (Figure 1B). Surface-enhanced Raman spectroscopy (SERS) is a surface-sensitive technique, allowing highly sensitive and selective structural analysis of adsorbed molecules. The discovery and development of SERS stimulate its fast-growing applications in both fundamental and practical research fields [3,4,5].

SERS was observed on a roughened silver (Ag) electrode [6,7,8], and the underlying mechanisms have been explored and discussed since then [9]. A variety of SERS-active materials are extensively developed, including noble metals, transition metals, and semiconducting materials [10]. Both electromagnetic and chemical enhancement mechanisms were proposed with increasing evidence obtained from experimental and theoretical studies [11,12]. According to the electromagnetic theory, surface plasmons occur upon stimulation by incident light where a collective oscillation of conduction band electrons exists at the interface of two materials [13]. When the light frequency matches the oscillation frequency of the electrons, known as surface plasmon resonance (SPR), an enhanced electrical field near the material surface will be induced, resulting in enhanced Raman scattering of adsorbates [14]. SPR-supported SERS is well accepted and is found capable of exhibiting ultra-sensitivity down to single molecular level at optimized experimental condition [15].

The plasmon theory alone could not explain all the SERS phenomena of varieties of molecules on varieties of substrates. When molecules have similar Raman cross sections, those chemically adsorbed on a material surface displayed larger Raman scattering enhancement. A chemical enhancement mechanism was proposed to explain such phenomenon. Charge transfer (CT) communication between an attached molecule and a certain nanomaterial is believed to contribute to SERS through the alteration of molecular polarizability [16]. The CT pathways from a material to a molecule or from a molecule to a material depend on the material, molecule, and laser energy [17]. With respect to a molecule adsorbed on a metal surface, photoinduced electrons can either be excited from the highest occupied molecular orbital (HOMO) of the molecule and transferred to the Fermi level of the metal, or excited from the Fermi level of the metal and transferred to the lowest unoccupied molecular orbital (LUMO) of the molecule [18,19]. The full valence band and empty conduction bands of semiconducting materials may function like the Fermi level of plasmonic materials as an electron donor or electron acceptor in a CT process [20]. The plasmon and CT mechanisms generally contribute together to overall SERS signals and their synergistic contribution of them at metal-semiconductor hetero nanostructures can exhibit extremely high Raman scattering enhancement [10].

SERS can provide detailed structural information of proteins under physiological condition without restriction in protein molecular mass and solubility. Since water has weak Raman scattering, SERS is a convenient tool for biological sample analysis. Two strategies are generally employed in SERS-based protein studies [21]. One aims to directly probe intrinsic protein structural information and the other obtains SERS of Raman labels to indirectly quantify or qualify proteins [5,22]. Only the former strategy can obtain intrinsic SERS fingerprints of proteins, which include vibrational information from amide groups, amino acids, and protein cofactors such as heme, flavins, etc. A protein cofactor can be selectively and sensitively detected by surface-enhanced resonance Raman spectroscopy (SERRS) where the excitation laser is in resonance with an electronic transition of the cofactor [23]. SERRS is powerful specifically for proteins in complex biological samples such as cells [24]. Structure characterization of proteins based on SERS is applicable in protein determination, protein redox processes, and protein–ligand interactions.

This review focuses on SERS-based structural characterization of proteins and its advances and applications. SERS-active nanomaterials, relevant synthetic approaches, strategies for protein-material assemblies, and how to improve the biocompatibility of SERS-active materials are firstly introduced. SERRS and SERS spectroscopy of proteins with and without cofactors is then discussed, and native and denatured protein fingerprints are compared in detail. Recent applications and advances of SERS spectroscopic proteins in biomarker detection, cell analysis, and pathogen discrimination are highlighted. Finally, SERS spectral reproducibility and limitations are critically discussed, and the review ends with a conclusion and a discussion of current challenges and perspectives.

## 2. SERS-Active Materials for Protein Immobilization

SERS was employed to investigate proteins in the late 1980s [25,26], where immunoglobulin G was adsorbed on a bare Ag electrode and vibrations from amino acid residues were obtained. Proteins with cofactors were also investigated and single molecular vibrations of a hemoglobin at the junction of two Ag nanoparticles were successfully probed by SERS [27]. Ag electrodes were used in the early studies, which were roughened usually by an anodization procedure. Ag and gold (Au) nanoparticles were successfully synthesized by Lee and Meisel [28], and commonly used soon after. Such metal colloids were prepared through the reduction of Ag nitrate by sodium citrate with a heating procedure. Metal island films fabricated by vacuum evaporation with an electron beam were also utilized for proteins [29]. Ag nanoparticles can aggregate mediated by proteins and thus display enhancement of Raman signals. Such substrates were mediated by proteins and used for protein identification [30,31,32]. Additionally, SERS-active semiconducting TiO_2_ and transition metal Ni films can be achieved by electrochemical anodization [33,34].

Proteins generally bind to noble surfaces via electrostatic interactions or noble metal-S bonds and the strong interactions may cause protein denaturation [32]. For example, conformational changes of cytochrome c (Cyt-c) were observed by SERRS on bare Ag nanoparticles [35,36]. Probing native structure of proteins is crucial for protein functional investigation. Thus, numerous efforts have been taken to improve the biocompatibility of noble nanoparticles. One useful strategy to improve their biocompatibility is to modify metal nanoparticles with spacer molecules (Figure 2). Ag electrodes are usually coated with self-assembled alkanethiols monolayers that carry charged head groups for protein soft immobilization [37,38]. Iodide-modified Ag nanoparticles allow label-free and highly sensitive detection of proteins. The coated iodide layer was found useful for preventing the direct interaction of the proteins with the metal surface, allowing protection of native protein structures [39]. The spacer molecules can also be polysaccharides such as chitosan, biocompatible for protein adsorption [40]. These strategies are all helpful for the improvement of material biocompatibility, but protein orientation is still uncontrollable. Random protein immobilization on SERS substrates usually cause poor spectral reproducibility (Figure 3A), making it challenging for SERS to probe protein functions without any extrinsic Raman labels.

Controlled immobilization and orientation of proteins on SERS-active materials are possible combined with metal-affinity chromatography (Figure 3A,B) [41]. Here, spacer molecules between proteins and SERS substrates were synthesized and optimized for capturing His-tagged proteins via nickel-imidazole coordination. This strategy balanced biocompatible protein immobilization and Raman scattering enhancement and is implicated in in situ probing protein functional versatility. This strategy was also employed to probe phosphorylated proteins via metal ion aluminum Al^3+^, which is ultrasensitive to discriminate a single-site phosphorylated protein [42] (Figure 3C). Controlled protein immobilization allows label-free fingerprinting amino acid residues near SERS-active substrates, and thus SERS can monitor local structure of proteins.

Beside noble metals, exploration of other SERS-active materials such as transition metals and semiconducting materials has attracted increasing interest. Semiconducting TiO_2_ nanostructures were found capable of enhancing Raman signal of cytochrome b_5_ (Cyt-b_5_) and without conformational change based on an electromagnetic enhancement mechanism [33]. Cyt-c can electrostatically adsorb on semiconducting K_2_Ti_6_O_13_ nanowires and its native conformation was observed. The K_2_Ti_6_O_13_ nanowires, a kind of wide band gap semiconductor with a multiple-layered structure consisting of eight uniform units of TiO_6_, were prepared by chemical modification of TiO_2_ nanoparticles. For the first time, photo-induced electron transfer from such nanowire to the protein was evidenced by Raman spectroscopy [43]. Note that the abilities of nickel nanomaterials for both Raman signal enhancement and an electron donor were discovered, which make it feasible to characterize Cyt c-cardiolipin interactions [34,44]. Other semiconducting materials such as quantum probes will be introduced in Section 6.2.

## 3. SERRS of Proteins

Protein cofactors can be selectively enhanced owing to resonance Raman effect. In combination of SERS and resonance Raman scattering, SERRS is ultrasensitive at a single-molecular level for some proteins with cofactors [34] or even without cofactors [45]. This section will introduce SERRS of hemes, flavins, and other chromophores constituted by amino acid residues.

### 3.1. Hemoproteins

Cyt-c is one of the hemoproteins that are most extensively investigated by SERRS. Complete resonance Raman spectra of Cyt-c was assigned by Hu et al. [46], according to which Cyt-c conformation information can be conveniently analyzed. Resonance spectroscopy is particularly powerful for characterizing heme proteins, since the vibrational modes are sensitive markers for the redox state, the spin and ligation pattern of the heme iron. Figure 4 shows the absorbance of oxidized Cyt-c and its resonance Raman fingerprints with the Soret-(407 nm) and Q-band (532 nm) excitation, respectively. Upon the Soret-band excitation, the resonance Raman spectrum is dominated by the totally symmetric modes A1g via the A-term enhancement mechanism. These bands lose intensity upon Q-band excitation and the non-totally symmetric modes B1g, A2g, and B2g gain intensity through the B-term enhancement mechanism [23]. Vibrational modes in the band region from 1300 to 1700 cm^–1^ are largely composed by C–C and C–N stretching vibrations of the porphyrin and the assignment details are listed in Table 1. The ν_4_ band is highly sensitive to the redox state of the Cyt-c, and together with ν_2_ and ν_3_ bands, these marker bands are sensitive to the spin and ligation pattern of the heme iron [47]. SERRS of Cyt-c on biocompatible materials displays almost the same fingerprint as its resonance Raman spectroscopy. However, direct contact of Cyt c with bare Ag nanoparticles would lead to broadened bands and frequency shifts due to protein denaturation (Table 1). Meanwhile, electron transfer would occur from Ag to the protein if a reduced Cyt c adsorbs on an Ag surface [35,48]. Combined with electrochemistry, protein structures and reaction dynamics are studied. Different oxidation, spin, and coordination states of Cyt-c were all clearly characterized by electrochemistry-SERRS (Figure 5A) [49].

Hemoproteins display similar SERRS fingerprints owing to the same porphyrin ring. Native structural of myoglobin (Mb) and Cyt-b_5_ were successfully probed on a magnetic Ag hybrid material coated with a biocompatible chitosan [40] and on a semiconducting TiO_2_ electrode [33]. As shown in Figure 5, only slight frequency shifts were observed from the SERRS of Mb and Cyt b_5_ with the same redox state. SERRS was also found capable of characterizing the slow and fast form of cytochrome c oxidase (heme a_3_), based on a marker band around 750 cm^−1^ [51].

### 3.2. Flavoproteins

Flavins as coenzymes play central roles in many oxidation–reduction reactions in living cells. Flavins include flavin monoucleotide (FMN), flavin dinucleotide (FAD), and riboflavin, and exist in three redox states: oxidized, semiquinone, and reduced states. Spiro et al. started to study SERS of FAD-containing glucose oxidase and riboflavin-proteins in 1984 [52], and resonance Raman spectra of FMN and FAD were investigated later [53]. SERS of protein-free flavin was investigated soon after [54]. SERRS of an FMN domain in nitric oxide synthase was observed on Ag nanoparticles coated with silica, and the spectra for oxidized and reduced FMN were obtained at different electrochemical potentials [55]. Importantly, the mode at around 1500 cm^−1^ was evidenced as a marker band for the redox states for FMN.

### 3.3. Other Proteins

Single-molecule SERRS was also observed in green fluorescent proteins, which are a type of proteins in which the chromophore is part of the protein without any cofactors [45]. The chromophore consists of three amino acid residues and is kept in place by a complex hydrogen-bonding network. The achievement of this study is implicated for structural dynamics of protein molecules at a single-molecule level. Resonance Raman spectra of peptides and proteins without intrinsic chromophores can be obtained by an excitation laser with a wavelength in the ultraviolet (UV) range [56], where proteins have an electronic absorption band. With a 229 nm laser excitation, UV-resonance Raman spectra would dominate by Tyr and Trp aromatic ring vibrations, and while with a deeper UV excitation at 206.5 nm, peptide bond amide vibrations would dominate in the overall spectra [57,58].

## 4. SERS of Proteins

Normal Raman spectra of proteins without any cofactors were comprehensively investigated and the relevant bands were assigned in detail [59,60]. Raman fingerprints generally exhibit vibrational information of amino acid residues, protein secondary structure (e.g., α-helical and β-sheet), and amide groups. For those proteins with cofactors but without resonance Raman effect when the applied laser energy does not match the electronic transition of the cofactors, SERS would exhibit bands from both amino acid residues and the cofactors [61]. SERS of such proteins often suffered from structural fluctuance owing strong interactions (through electrostatic interactions or through metal-S bonds) with bare noble metals. SERS fingerprints of proteins may significantly differ from their normal Raman spectra either due to protein denaturation or random orientation [41]. SERS fingerprints of proteins are versatile and are highly dependent on their intrinsic spatial structures, their orientations on SERS-active materials, and the interaction types between proteins and SERS-active materials. Figure 6A shows SERS spectra of five proteins including two hemoproteins, hemoglobin, and Cyt-c. Those heme-dominated SERRS spectra are very similar as discussed above, but SERS profiles of the two hemoproteins are significantly different. The SERS of three other proteins are also distinguishable although they only have peptides. One point that should be mentioned here is SERS spectral reproducibility of these proteins sandwiched between Ag nanoparticles and measured in solution is much better that those dried samples [31]. However, these SERS profiles reflected denatured protein structures.

Studies have demonstrated that the feasibility of SERS to probe native protein structures [39]. Here, bare Ag surface coated with iodide ions prevented strong interactions of proteins with Ag surfaces, preserving native structure of proteins with positively charged proteins. Both positively and negatively charged proteins were successfully probed by a further study, in which aluminum and iodide ions were added [63]. Figure 6B shows SERS spectra of native proteins. With respect to the same protein, lysozyme at the two panel of Figure 6, band wavenumbers were significantly different, indicating the protein structure on the two Ag aggregates are totally different. Note that unlike those denatured proteins as shown in the left panel of Figure 6, the two hemoproteins myoglobin and catalase displayed more similar SERS profiles as the three other proteins in the right panel of Figure 6. Proteins can be discriminated by the overall spectral differences and the quantity of aromatic amino acid residues in these proteins can be compared by relative band intensities.

## 5. SERRS/SERS of Protein-Ligand Complex

Overall SERS of proteins can be probed in mobile Ag nanoparticle colloids as discussed above. However, it is still a big challenge for this strategy to in situ probe protein–ligand interactions, because ligands can also interact with Ag nanoparticles. Alternatively, it is reasonable to fix proteins on immobilized substrates, and subsequently probe their structural alteration during the binding process of protein ligands. According to the electromagnetic enhancement theory, only those amino acid residues that are very close to the SERS-active surface, their Raman signals can be remarkably enhanced [64,65]. Therefore, Raman scattering of only a small domain (a cofactor for SERRS or a specific protein domain containing amino acid residues for SERS) of a large-size protein would be enhanced.

SERRS was successfully employed to probe the interaction of reduced Cyt c with cardiolipin (CL) (Figure 7A) [34]. The electron donor property of the Ni can ensure the reduced state of Cyt-c in air, thus to eliminate the autoxidation of the reduced Cytc before CL binding under aerobic conditions. The ν_4_ band frequency in the Cyt-c_(red)_–CL complex upshifts about 3 cm^–1^ (1376 cm^–1^) compared to that from the Cyt-c_(ox)_–CL complex (1373 cm^–1^) (Figure 7B), indicating a more significant conformation, which was further confirmed by molecular dynamic simulation (Figure 7C). These spectral changes probably reflected a conformational transition to a non-native state in which the axial Met-80 ligand is replaced by a His ligand [66,67]. Remarkably, the roughened Ni surface is effective for in situ monitoring Cyt-c released from apoptotic mitochondria by SERRS. Additionally, it is also possible for SERS to monitor protein–drug interactions on a cross-linker modified Ag substrate. A model protein α-fetoprotein and a drug all-trans-retinoic acid was studied on an iminodiacetic acid-modified Ag film. The interaction of all-trans-retinoic acid with α-fetoprotein was evidenced based on the shifted, newly emerged, and disappeared SERS bands of the α-fetoprotein [41]. However, further optimization of the experimental conditions is needed for improving the spectral reproducibility and at the current stage, this strategy is only applicable for small proteins.

## 6. Applications in Biology and Biomedicine

SERS fingerprints are specific for proteins owing to their primary and spatial structures. In addition to fundamental studies, SERS is helpful for the determination or discrimination of proteins either free or integrated in cell membranes in practical applications in biology and biomedicine [68]. Structural information of proteins in cytosol, in serum, or at pathogen walls are of significance for mechanical exploration in biological processes and clinical diagnosis.

### 6.1. Protein Biomarkers

High selectivity of SERRS for protein cofactors enables high sensitivity of SERRS for the detection of those protein biomarkers with cofactors. For example, Mb is a biomarker for acute myocardial infarction, high sensitivity for the detection of myoglobin based on its intrinsic SERRS was achieved by a 3D Ag anisotropic nanopinetree array [69]. Moreover, multiplexed protein analysis of three hemoproteins was achieved by an inverse opal photonic crystal hydrogel [70]. It is also possible for those protein biomarkers without cofactors or chromophores to be detected by SERS [71], but it is still challenging for SERS to discriminate one protein from a complicated biological sample. Thus, great efforts have been taken to design functional materials for selectively capturing target proteins. Molecularly imprinted polymers (MIP)-SERS has been proven effective in improving the selectivity of SERS for protein biomarkers. Transferrin and BSA were selectively detected by MIP-SERS nanosensors [72,73]. In the study for transferrin, characteristic SERS peaks assigned to the vibrations of amide I (1626 cm^−1^) and amide III (1221 cm^−1^) of α-helix structure, and other abundant bands due to amino acid residues were observed. In another promising way, protein biomarkers could be captured by their specific aptamers or antibodies and subsequently detected by SERS through the vibrational modes of amide III at 1300 cm^−1^, amide II at 1550 cm^−1^ and amino acid residues [74,75,76,77].

### 6.2. Proteins in Tissues

SERS of proteins in tissues are detectable by directly dropping noble metal nanoparticles on tissues or mixing tissues with a nanoparticle solution, and nanoparticle coating will form at the surface of the tissues [78,79,80]. Alternatively, tissues can be placed on SERS-active substrates and SERS collected from the top of the issues [81]. Such a non-specific approach is mostly adapted for discriminating normal and diseased tissues by differential SERS spectra including vibrations from proteins and other species. Moreover, SERS can also probe protein aggregation in tissues by in situ formation of metal nanoparticles with Aβ monomers and fibrils as templates. In such case, these nanoparticles in turn served as an effective substrate for enhancing Raman signals of Aβ monomer and fibril. On the basis of SERS characteristic bands of Aβ monomer and fibrils, real-time Aβ aggregation process in neurons and brain tissues was successfully monitored by ratiometric SERS [82]. Note that the developed SERS platform provided remarkable spatial resolution, allowing intracellular mapping of Aβ_40_ aggregation at the single cell level. Moreover, nanoparticles can be injected directly into tissues for diagnosis of cancers based on SERS of components in tissues such as proteins [83]. Here, SERS has been proven capable of serving as promising indicators for identifying the primary tumors in different stages.

In situ formation of nanomaterials and nanoparticle injection are both effective for comprehensive contact of tissue components with SERS-active materials. However, owing to the complexity of tissues, accurate discrimination of normal and cancer tissues usually needs the aid of chemometric approaches [84].

### 6.3. Intracellular and Extracellular Proteins

An intracellular protein would form protein corona around metal nanoparticles [85]. Cellular uptake and intracellular dynamics of gold nanoparticles were investigated and SERS spectral fingerprints supported that the composition of the protein corona formed on metal particles [86]. Extracellular vibrational spectroscopy produces spectra which can be considered as a fingerprint of the cell. A single-cell Raman spectrum generally contains Raman shifts including nucleic acids, proteins, carbohydrates, and lipids. A microfluidic device for continuous single cells analysis was developed and SERS collected from cells on nanodimers for Raman signal enhancement [87]. Both amino acid residues and Amide III vibrations of proteins were observed, and this microfluidic device will potentially become a fast cell diagnostic tool. Although it is not easy to identify SERS bands due to multiple components, SERS combined with chemometric analysis or machine learning can be further used to distinguish between normal cells and cancer cells [88,89,90].

Use of unique properties of quantum materials for SERS biosensing is an emerging and promising field [91]. High surface area, good crystallinity, and biocompatibility of quantum materials allow for multiple applications of sensing and diagnostics [92]. Recent studies showed that non plasmonic semiconductor quantum probes decorated on a 3D platform enable a single-cell-level detection and the discrimination of cancerous and non-cancerous cells along with biomolecular sensing in vitro [93]. Intense SERS of proteins observed from such probes were considered to originate from vibrionic coupling, charge transfer resonance, and surface plasmon resonance (Figure 8). Such noble-metal-free methodology was also proven helpful for unveiling the “biochemical fingerprint” of substantial components of cells by a Si@SiO_2_ quantum probe [94].

### 6.4. Proteins of Pathogens

SERS has been proved to be feasible and powerful to detect pathogens such as bacteria and viruses [95,96], or detect pathogen infection through related metabolomics [97,98]. Generally, protein SERS is either collected from the outside of pathogen walls or within the plasma inside the wall through the interactions of proteins with metal nanostructures such as colloid and roughened films. Ag colloid can be synthesized both inside and outside bacteria by in situ reduction [99]. Interestingly, SERS spectra obtained from Ag colloid coated bacteria are dominated by spectral features of flavins incorporated into the cell wall [100,101]. In contrast with those obtained from Ag-coated bacteria, SERS intensities observed inside bacteria were rather weak and most bands were assigned to amino acids [101]. Alternatively, bacteria were often directly adsorbed on roughened metal films and they can specifically be captured by antibodies, aptamers, antibiotics, or bacteriophages [95]. Vibrational information proteins from bacteria for example, Amide III, υ(COO-) and δ(C-H) were observed [102,103].

SERS has the potential to identify bacterial species by unique molecular compositions. However, it is still a big challenge to obtain consistent and clear spectra due to many overlapping peak sources such as proteins in cell walls. Multivariate data analysis methods such as principal component analysis (PCA) have been applied for bacterial classification [99]. Furthermore, direct detection of bacteria in solutions is more challenging because bacterial signals are substantially overlapped by those from the media itself or are masked by an additional fluorescence background. Combining of SERS with machine leaning is a promising strategy for bacterial classification. Studies show that deep learning has been proven effective for rapid identification of pathogenic bacteria [104]. Deep learning is an artificial intelligence method for hierarchical extraction of features using multiple layers with trainable parameters (Figure 9A). In a recent study, separation-free bacterial identification was achieved on Ag-coated nanopillar arrays combined with deep learning (Figure 7B) [105]. With outstanding classification accuracies up to 98%, this method allows for the detection of bacteria in arbitrary media with short data acquisition times and small amounts of training data.

SERS has also been proven useful for virus detection and it has attracted increasing interest in recent years [106]. In recent years, during the COVID-19 pandemic, rapid and label-free detection methods of SARS-CoV-2 were developed by the identification of the characteristic peaks of their outer membrane proteins [107,108]. It is worth mentioning that an ultrasensitive SERS biosensor for the detection SARS-CoV-2 virus at single-virus level was achieved as shown in Figure 10. Here, an ACE2-functionalized hierarchical gold nanoneedles array was used as a SERS sensor for selective capturing and culture-free identification of SARS-CoV-2 in the contaminated water. The high affinity of ACE2 with S protein and the as-designed “virus-traps” nanoforest synergistically improved the accuracy of such SERS sensors. The characterized SERS bands for the S protein of SARS-CoV-2 at 568, 884, and 1296 cm^−1^ were observed, which are attributed to the vibrations of Amide V, Trp, and Amide III. This strategy can be used to quickly establish the identification standard based on their SERS spectra of the emerging coronaviruses and machine-learning techniques.

## 7. SERS Spectral Reproducibility and Limitations

Reproducibility of protein SERS includes the spectral reproducibility in the overall band intensities, relative intensities, and frequency shifts [3]. As discussed above, how to obtain a reproducible SERS of a target protein especially for those without any cofactor or an intrinsic chromophore, is a key point for its structural characterization. For purified proteins, it is helpful for sandwiching and detecting them with colloidal metal nanoparticles, resulting in reproducible SERS from native proteins [63]. However, it is difficult for this strategy to probe a certain protein structure among protein mixtures or more complicated biological samples, and it is also challenging to monitor protein structural fluctuation upon ligand binding. Placing proteins on a nanostructured surface with high biocompatibility is an alternative strategy to address these issues. In this case, random orientations often cause poor spectral reproducibility, and site-specific protein immobilization is expected to make protein orientations controllable. However, this strategy currently only works for small proteins and further optimization is needed [110].

SERS-based identification of membrane-proteins integrated in cellular, virtual, or bacterial walls also suffered from spectral fluctuation if the targets were randomly adsorbed on SERS-active materials. Functionalization of nanomaterials with specific antibodies or aptamers would be helpful for controlled protein capturing [109]. However, the majority of proteins without chromophores have small Raman cross-sections and thus exhibit relatively weak signals with band overlaps. Therefore, it is highly required for developing biocompatible, highly sensitive, and selective SERS-active materials for adhesion of such biological samples. For those proteins within cells, SERS fingerprints more easily suffer from fluctuating due to dynamic movements.

In addition, the laser power and exposure time also affect SERS reproducibility. Measurements with high laser power or longer laser exposure times yield improved signal-to-noise ratios, but they may also cause sample damage and the changes in spectral intensity and fingerprints owing to photobleaching and photo-induced chemical reactions [13]. Thus, how to obtain intense protein SERS signals and meanwhile protect its intact structure and biological function from laser-induced side effects should carefully be considered during experimental procedures.

## 8. Conclusions and Outlook

This review introduced and discussed SERS for protein structural characterization, including SERS-active materials and how to improve their biocompatibility for protein immobilization, SERS/SERRS fingerprints of native and denatured proteins, and its applications in protein biomarker quantification, cell detection, and pathogen discrimination. SERS spectral reproducibility and limitations were highlighted on the basis of a deep understanding of this research field. The latest and remarkable achievements introduced in this review suggest the great potential of SERS for label-free protein analysis. However, challenges still exist with respect to spectral overlaps and reproducibility, materials biocompatibility and toxicity, and identification accuracy. With the development of material science and Raman instrument, SERS has experienced significant growth in protein studies for both fundamental and applied studies. Meanwhile, novel methods that can improve the current limitations are highly desirable, and the promising directions are outlooked in the following.

SERS combined with computational chemistry methods is promising for protein structural explanation. Density functional theory calculations is useful for protein band assignments, and molecular dynamics simulation is helpful for interpreting the underling mechanism of SERS spectroscopic alteration. Moreover, integrating machine learning with SERS is a promising way to achieve highly accuracy of protein analysis in complex matrices. Further exploration of SERS combined with computational chemistry will significantly improve the reliability and multiplexing ability of SERS-based protein analysis. On the other hand, it is of significance for in vitro protein analysis to develop novel materials with better biocompatibility and higher SERS activity. To this end, non-plasmonic semiconductor quantum probes are promising. Finally, label-free, non-invasive SERS cell imaging with a higher spatial resolution at the level of small organelles could be another promising direction, which is beneficial for revealing the mechanisms of signaling transduction pathways in cells.

## Figures and Tables

**Figure 1 ijms-23-13868-f001:**
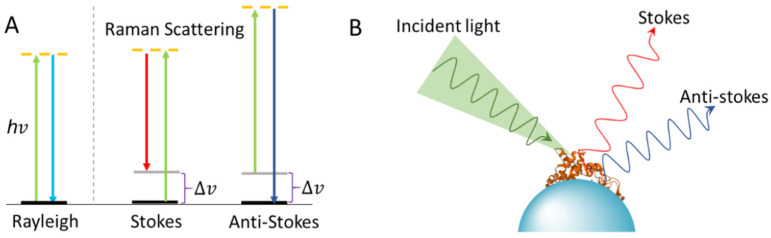
Light scattering and SERS. (**A**) Energy changes of Rayleigh, Stokes, and anti-Stokes scattering. (**B**) Enhanced Raman scattering of a protein adsorbed onto a nanoparticle.

**Figure 2 ijms-23-13868-f002:**
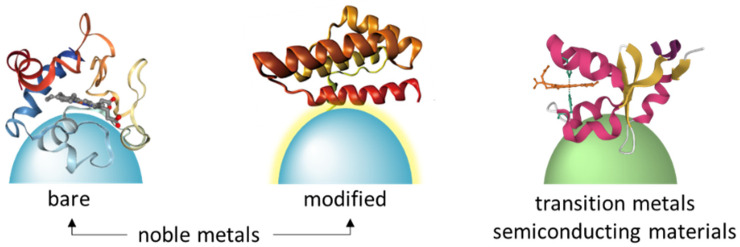
Immobilization of proteins onto SERS-materials.

**Figure 3 ijms-23-13868-f003:**
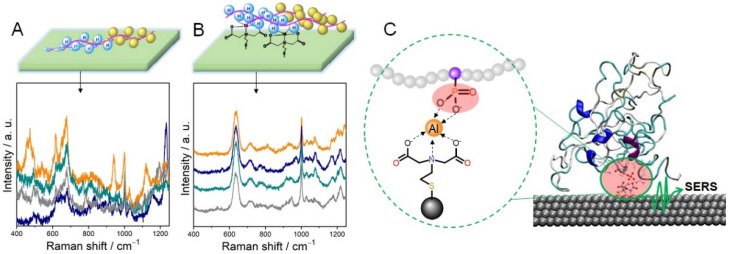
Iminodiacetic acid-modified Ag nanoparticles for His-tagged free (**A**), His-tagged (**B**), adapted with permission from [41]. Copyright [2019] American Chemical Society; (**C**) phosphorylated protein determination, adapted with permission from [42]. Copyright [2021] Elsevier.

**Figure 4 ijms-23-13868-f004:**
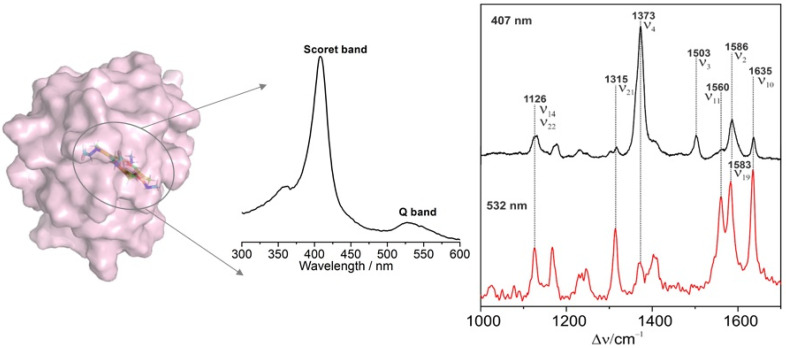
Absorbance (blue) and resonance Raman spectra of oxidized Cyt-c with 407 (black) and 532 nm (red) excitation, respectively [50].

**Figure 5 ijms-23-13868-f005:**
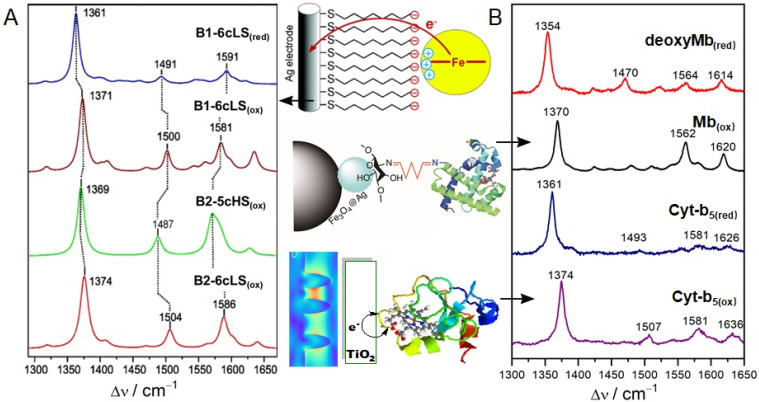
SERRS of hemoproteins including Cyt-c, Mb, and Cyt-b_5_. (**A**) SERRS of different oxidation, spin, and coordination states of Cyt-c immobilized on a CO_2_-SAM Ag electrode measured with 413 nm excitation. Reprinted with permission from [49]. Copyright [2004] American Chemical Society. (**B**) (Black) SERRS of deoxyMb_(red)_ and Mb_(ox)_ on a magnetic-Ag hybrid material (adapted with permission from [40]. Copyright [2013] American Chemical Society), and (purple) two redox states of Cyt-b_5_ on a TiO_2_ electrode (adapted with permission from [33]. Copyright [2013] John Wiley and Sons).

**Figure 6 ijms-23-13868-f006:**
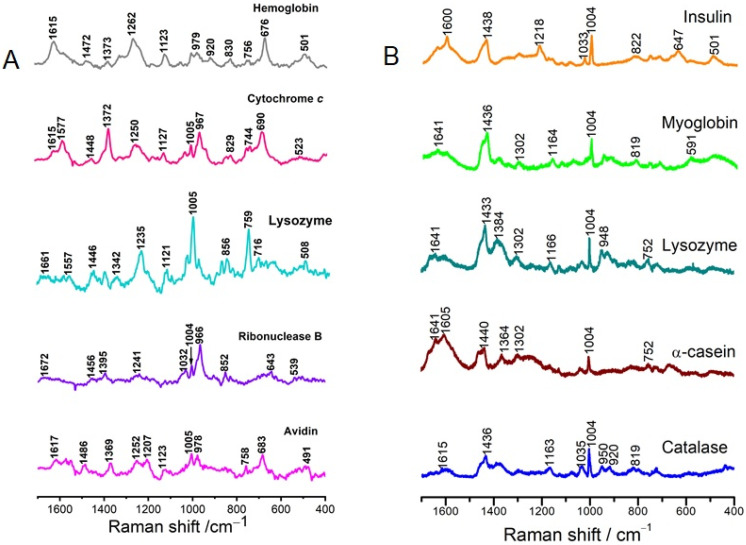
SERS of proteins collected from acidified sulfate-induced Ag aggregates with a 785 nm laser excitation (**A**), adapted with permission from [62]. Copyright [2009] American Chemical Society and (**B**) from aluminum and iodide ions-induced Ag aggregates with a 633 nm laser excitation, adapted with permission from [63]. Copyright [2020] American Chemical Society.

**Figure 7 ijms-23-13868-f007:**
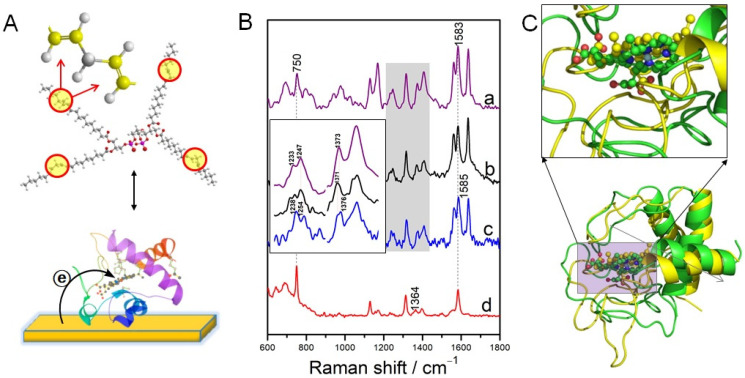
RR/SERS spectra of Cyt c/Cyt c-CL complexes. (**A**) Structure of CL and Cyt-c; (**B**) RR spectra of Cytc_(ox)_-CL complexes (a) and (b) Cytc_(ox)_; SERS spectra of Cyt c_(ox)_ on the Ni substrate before (d) and (c) after adding CL liposomes; (**C**) molecular dymanic simulation results of the binding of Cyt c to CL, Cyt-c(ox) (yellow) and Cyt c(red) (green). Adapted with permission from [34]. Copyright [2019] John Wiley and Sons.

**Figure 8 ijms-23-13868-f008:**
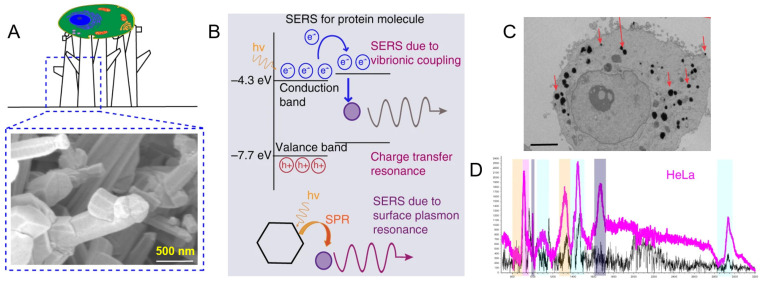
ZnO quantum probes decorated on a 3D platform for a single-cell-level detection [93] (**A**) Three-dimensional tetrapod morphology of the nano-dendrites allowed cells adhesion and proliferation; (**B**) cytoplasm was benefited by the cumulative resonances of charge transfer as well as surface plasmon providing enhanced signals for proteins; (**C**) a TEM image of a cell showing cellular uptake and internalization of the quantum probe, Scale bar = 5 µm; (**D**) enhanced SERS signal for HeLa cells.

**Figure 9 ijms-23-13868-f009:**
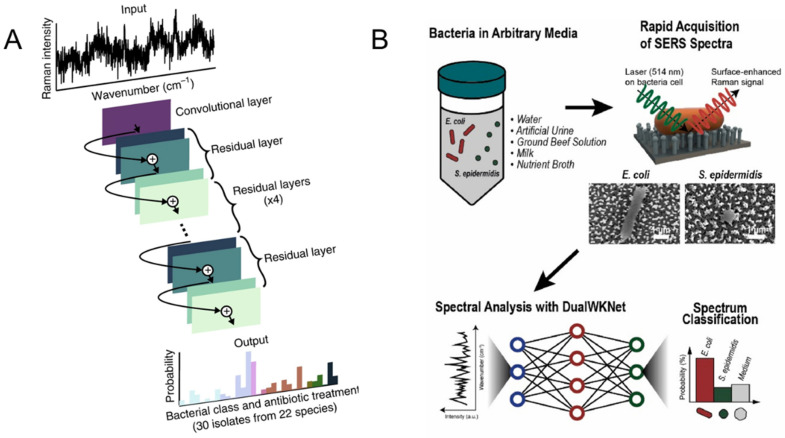
SERS combined with deep learning. (**A**) Using a one-dimensional residual network, low-signal Raman spectra are classified, which are then grouped by empiric antibiotic treatment [104]; (**B**) schematics of the general process of Raman data collection and analysis where a single spectrum is attained from a single cell and classified via deep learning, adapted with permission from [105]. Copyright [2022] Elsevier.

**Figure 10 ijms-23-13868-f010:**
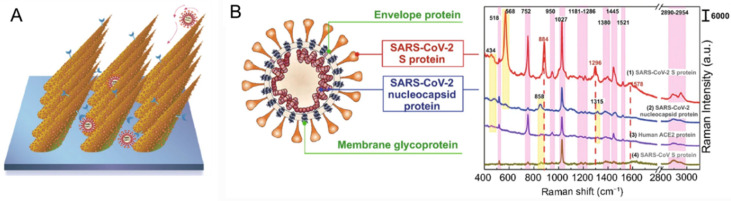
Gold-nanoneedles array and SERS spectra of viral protein. (**A**) Schematics of “virus-traps” nanoforest composed of tilted gold-nanoneedles array. (**B**) Structure schematics of SARS-CoV-2 (**left**), and SERS spectra (**right**) of SARS-CoV-2 S protein and nucleocapsid protein, SARS-CoV S protein, and Human ACE2 protein [109].

**Table 1 ijms-23-13868-t001:** Band assignments of resonance Raman and denatured SERRS of Cyt-c, Cyt-b_5_ and Mb [40,44,46,48].

Frequencies						Assignments
SERS			Raman			
Mb	Cyt-b_5_	Cyt-c	Cyt-c	Cyt-b_5_	Mb	
1621	1630	1640	1635	1626	1620	ν_10_ ν(CαCm)asym
1565	1579	1585	1583	1581	1562	ν_2_ ν(CβCβ)
		1567	1560			ν_11_ ν(CβCβ)
1489	1492	1505	1500	1493	1479	ν_3_ ν(CαCm)sym
		1406	1401			ν_29_ ν(pyr quarter-ring)
1370	1374	1374	1371	1374	1370	ν_4_ ν(pyr half-ring)sym
		1317	1314			ν_21_ δ(CmH)
		1170	1168			ν_30_ ν(pyr half-ring)sym
		1130	1129			ν_22_ ν(pyr half-ring)sym

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
