# Peer review of "Label-Free Surface-Enhanced Raman Spectroscopic Analysis of Proteins: Advances and Applications"

_ijms, 2022, doi:10.3390/ijms232213868_

Round 1

Reviewer 1 Report

Very well written review encompassing major applications. Minor spell check is recommended along with simplifying the sentences. I feel some of the sentences are long and complicated. Breaking them into simpler ones will ease the reader.

My only suggestion to the authors is that they should discuss the tissue-based applications and the current challenges/ unforeseen challenges that might creep up during the studies. In general, the widescale applicability of SERS in tissue based studies.

Author Response

Your suggestion about the tissue-based applications is greatly appreciated. A new Section 6.2 has been accordingly added in Page 9 (highlighted in red). Additionally, some long sentences in the text have been modified to avoid misunderstanding.

Reviewer 2 Report

In the submitted manuscript, Cai et al. described the applications of SERS/SERRS in the analysis of the proteins’ structures, their interactions with ligands, their adsorption of different materials and SERS application for biomarkers’ detection. Additionally, the authors gave readers some limitations linked with using SERS protein analysis as well as the perspective remarks related to future research work.

The article is interesting and provides up-to-date information, confirmed by 41% of the latest cited references.

The reviewer suggests that this manuscript be accepted for publication after minor improvements such as:

- insert the year in reference 79.

Author Response

Thank you very much for all your comment. The year in Reference 79 has been added.

Reviewer 3 Report

Review on label-free surface-enhanced Raman spectroscopy analysis of proteins by Han et. al. is informative and will help readers learn about the technique and its biological applications. Following comments might help increase clarity.

A. Line 87 = critical to critically

B. Replace "highly Raman signal enhancement" to "enhancement of Raman signal"

C. Figure 3 = In figure caption, please mention details about A, B and C sub figures.

D. Line 155 = "one of" to "one of the"

E. Figure 4 = The text on figure is not clear due to size and resolution. The black and blue colors for spectra are not distinguishable.

F. Lines 218 - 221 = Could authors prepare a table of SERS band assignments for proteins without cofactors similar to Table 1? This will help readers learn nuances of SERS.

G. Lines 221-224 = Please reframe the sentences as they are not very clear.

H. Line 227 = Replace "seldom" to "random"?

I. Line 228 = "dependend" to dependent"

J Line 238 = "Studies demonstrated" to "Studies have demonstrated" 

K. Lines 273-279 = Please reframe for better clarity.

L. Line 284 = Replace "similation" to "simulation"

M. Lines 290-292 = Please reframe for better clarity.

N. Line 304 = capture to capturing

O. Line 343 = "has been proved" to "has proved"

P. Line 363 = Authors have mentioned clearly bacterial classification. Do they mean clarifying bacterial classification?

Q. Line 410 = "targets randomly" to "targets were randomly"

Author Response

All your comments and suggestions are very much appreciated.

The typos mentioned in (A, B, D, H, I, J, L, N, O, and Q) were corrected, and the confusing sentences mentioned in (G, K, and M) are reframed.

  1. The caption of Figure 3 was modified with details about the labels.
  2. A new Figure 4 with a high resolution was added.
  3. Since SERS of those proteins without cofactors significantly depend on SERS-active materials and interactions between the materials with the proteins, SERS spectra of one protein obtained in different labs usually display different profiles. Thus, it is hard to provide a table with SERS band assignments of such proteins. As mentioned on Page 6, normal Raman spectra of proteins without any cofactors were comprehensively investigated and the relevant bands were assigned in detail [62, 63]. Currently, one can assign SERS bands of such proteins by referring to the corresponding Raman bands.
  4. Here “clear” was deleted for clarity.
